# Could Reducing Body Fatness Reduce the Risk of Aggressive Prostate Cancer via the Insulin Signalling Pathway? A Systematic Review of the Mechanistic Pathway

**DOI:** 10.3390/metabo11110726

**Published:** 2021-10-23

**Authors:** Rachel James, Olympia Dimopoulou, Richard M. Martin, Claire M. Perks, Claire Kelly, Louise Mathias, Stefan Brugger, Julian P. T. Higgins, Sarah J. Lewis

**Affiliations:** 1Department of Population Health Sciences, Bristol Medical School, University of Bristol, Bristol BS8 2BN, UK; rachel_symes@hotmail.co.uk (R.J.); Olympia.Dimopoulou@bristol.ac.uk (O.D.); richard.martin@bristol.ac.uk (R.M.M.); Claire.Kelly@bristol.ac.uk (C.K.); Louisemathias@hotmail.com (L.M.); Stefan.Brugger@bristol.ac.uk (S.B.); Julian.Higgins@bristol.ac.uk (J.P.T.H.); 2Medical Research Council Integrative Epidemiology Unit, University of Bristol, Bristol BS8 2BN, UK; 3IGF & Metabolic Endocrinology Group, Translational Health Sciences, Bristol Medical School, University of Bristol, Bristol BS10 5NB, UK; claire.m.perks@bristol.ac.uk

**Keywords:** body fatness, prostate cancer, biomarker, insulin, insulin signalling, mechanisms, systematic review

## Abstract

Excess body weight is thought to increase the risk of aggressive prostate cancer (PCa), although the biological mechanism is currently unclear. Body fatness is positively associated with a diminished cellular response to insulin and biomarkers of insulin signalling have been positively associated with PCa risk. We carried out a two-pronged systematic review of (a) the effect of reducing body fatness on insulin biomarker levels and (b) the effect of insulin biomarkers on PCa risk, to determine whether a reduction in body fatness could reduce PCa risk via effects on the insulin signalling pathway. We identified seven eligible randomised controlled trials of interventions designed to reduce body fatness which measured insulin biomarkers as an outcome, and six eligible prospective observational studies of insulin biomarkers and PCa risk. We found some evidence that a reduction in body fatness improved insulin sensitivity although our confidence in this evidence was low based on GRADE (Grading of Recommendations, Assessment, Development and Evaluations). We were unable to reach any conclusions on the effect of insulin sensitivity on PCa risk from the few studies included in our systematic review. A reduction in body fatness may reduce PCa risk via insulin signalling, but more high-quality evidence is needed before any conclusions can be reached regarding PCa.

## 1. Introduction

Prostate cancer (PCa) is the leading cause of morbidity and mortality in men worldwide [1]. In 2018, there were 1.28 million newly diagnosed PCas, making the disease the second most commonly occurring cancer in men [2]. Despite the importance of the disease, little is understood about the causes of PCa. Established risk factors of the disease—age, race and family history [3]—are not modifiable and, therefore, a principal focus of current research is to advance our understanding of modifiable risk factors.

Excess body weight, resulting from imbalances in diet and physical activity, has been implicated as a risk factor for the incidence and progression of several cancers including PCa [4,5]. In 2012, overweight and obesity, characterised by excess body fat, w estimated to contribute to 3.9% of all incident cancers, a figure expected to rise in the next few decades given current trends in this risk factor [6]. Whilst the association with localised PCa remains inconsistent, evidence has shown that body fatness is positively associated with aggressive PCa, PCa recurrence and survival [7,8,9]. In a meta-analysis of 31 cohort and 25 case–control studies, a higher body mass index (BMI) was associated with a greater risk of PCa (overall relative risk (RR): 1.05 per 5kg/m^2^ increment; 95% confidence interval (CI) 1.01–1.08). In the same study, the relative risk was stronger for advanced (RR 1.12 per 5 kg/m^2^ increment 95% CI 1.01–1.23) PCa and weaker and inverse for localised PCa (RR 0.96 per 5 kg/m^2^ increment, 95% CI 0.89–1.03) compared with controls [10].

With the increasing worldwide prevalence of overweight and obesity in men [11], a better understanding of the molecular mechanisms that underpin the relationship between excess body fatness and PCa is warranted. Understanding such mechanisms may help to promote preventative lifestyle strategies to reduce the incidence and progression of PCa and may highlight targets for intervention, treatment and risk monitoring [12].

Insulin is a hormone made and secreted by the pancreas that plays a key role in the regulation of blood glucose levels. Insulin binds to receptors on cell membranes, initiating a cascade of reactions that activate specific signalling pathways within the cell. This results in cellular uptake of glucose, promoting cell growth and inhibiting apoptosis [13].

Body fatness is positively associated with increasing blood insulin levels [14,15,16], causing a diminished cellular response to insulin termed insulin resistance. The reduced cellular response to insulin mainly occurs in insulin-target tissues, such as muscle, fat and liver. Cancer cells do not become insulin resistant, insulin resistance leads to the body producing more insulin to compensate, which, combined with an overexpression of insulin receptors on cancer cells, may lead to the promotion of cancer initiation and progression [17]. Insulin resistance is common among overweight and obese individuals and leads to hyperinsulinemia, where insulin levels are chronically elevated relative to blood glucose. In some cases, insulin resistance leads to the development of type 2 diabetes mellitus (T2DM), a metabolic disease that is positively associated with body fatness [18,19,20].

Evidence suggests that insulin and biomarkers of higher circulating insulin, including C-peptide, are positively associated with PCa risk [21]. A diagnosis of T2DM, however, seems to be protective [22,23], which may be due to the potential anti-neoplastic biological effects of diabetic medications, such as metformin [24]. Alteration of the insulin signalling pathway is therefore a plausible mechanism underlying the link between body fatness and PCa.

Whilst there is evidence that excess body fatness is a risk factor for aggressive PCa [10], it is not clear whether reducing body fatness will reduce cancer progression, and whether it does this via effects on insulin sensitivity. Follow-up periods for studies examining interventions aiming to reduce body fatness are insufficiently long to assess the impact on prostate cancer diagnoses [15]. Recently, an International Agency for Research on Cancer (IARC) working group produced a framework to assess evidence for the effect of interventions on cancer risk for their cancer prevention handbooks. The framework uses a two-step approach, where two different sets of studies are combined: studies in set 1 examine the effect of the intervention on a mechanistic intermediate and set 2 examines the same intermediate with respect to cancer risk [25]. This is a similar approach to the one we took when developing the World Cancer Research Fund (WCRF)/Bristol Methodology to assess the mechanism between an exposure and cancer risk [12]. In this methodology the first stage is designed to identify mechanisms underpinning a specific exposure–disease relationship and prioritise these mechanisms using specifically designed text mining tools. The second stage is a targeted systematic review of studies of the exposure of interest and intermediate phenotypes relating to the mechanism of interest, and a separate systematic review of studies of the same intermediate phenotypes and outcome of interest. In this systematic review, we used the second stage of this methodology [12] to assess the evidence for: (1) the effect of intervening to reduce body fatness on biomarkers in the insulin signalling pathway; and (2) the effect of biomarkers in the insuling signalling pathway on prostate cancer risk.

## 2. Material and Methods

### 2.1. PICO Questions

Our objective was to systematically review and synthesise evidence from studies investigating whether changing body fatness might impact on PCa risk via the insulin signalling pathway; in doing so, we investigated the insulin signalling pathway as a potential mechanistic link between body fatness and PCa incidence or progression.

Relevant studies which contributed evidence for mechanistic links were those that reported an intermediate phenotype (here the insulin signalling pathway) as either an outcome or an exposure. Defined using PICO/PECO (population, intervention/exposure, control, and outcomes), we separately identified studies which assessed: (1) the effects of dietary interventions (I) to reduce body fatness in adult males (P) on insulin outcomes (O) compared to controls (C); and (2) whether insulin signalling (I/E) in adult males (P) impacted on PCa outcomes (O) compared to controls (C). Our interest was primarily in humans so we prioritised studies in humans and a priori decided to only review animal studies where the former were absent.

### 2.2. Standards of Reporting

This systematic review and meta-analysis followed the WCRF International/University of Bristol methodological framework [12]. The review was registered in the PROSPERO International Prospective Register of Systematic Reviews (CRD42020196064) and reported according to the Preferred Reporting Items for Systematic review and Meta-Analysis Protocols (PRISMA-P) checklist. As all analyses were based on the published results of previous studies, no ethical approvals or patient consent were required.

### 2.3. Inclusion and Exclusion Criteria

#### 2.3.1. Body Fatness–Insulin Signalling-Specific Criteria

We included experimental studies examining the effects of dietary intake interventions to change body fatness on the insulin signalling pathway. We excluded studies where the only intervention had a physical activity component, as physical activity is likely to have an independent effect on insulin signalling. Randomised controlled trials (RCTs) with human adult participants were eligible for inclusion, with no restriction on race or nationality. Since only males can develop prostate cancer, we only included studies with male subjects analysed separately. Studies which included a high proportion of participants with a diagnosis of type 1 or 2 DM at baseline were excluded, although we included population-based studies which did not exclude participants on the basis of diabetes.

Our exposure of interest was a change in body fatness measured by indirect methods including anthropometric biomarkers (BMI, waist circumference, waist-to-hip ratio (WHR), crude weight and skinfold thickness) and bioelectrical impedance analysis (BIA), or direct methods including densitometry, computed X-ray tomography (CT), magnetic resonance imaging (MRI), and dual-energy X-ray absorptiometry (DXA).

Outcomes of interest were serum or plasma insulin or biomarkers and surrogate indices of the insulin signalling pathway and insulin resistance (fasting glucose, C-peptide, pro-insulin, homeostatic model assessment for insulin resistance (HOMA-IR), homeostatic model assessment for insulin sensitivity (HOMA-S), glycated hemoglobin (HbA1c) and quantitative insulin sensitivity check index (QUICKI)).

#### 2.3.2. Insulin Signalling–PCa-Specific Criteria

We included studies examining the association between the insulin signalling pathway and PCa outcomes. Only RCTs or prospective observational studies were eligible. To reduce the possibility of the results being affected by reverse causation we only included studies in which exposure was measured at least two years (or with a study mean/median of at least 5 years) before outcomes were counted. Cross-sectional and retrospective study designs were excluded. Commentaries, editorials and conference proceedings and studies published only as protocols were excluded.

Studies of human adult males (aged 18 years or over) with no restriction on race or nationality were eligible for inclusion. Men with a PCa diagnosis at baseline were excluded, with the exception of studies examining cancer progression as the outcome. Within such studies, men undergoing androgen deprivation therapy (ADT) for PCa were excluded. ADT increases the risk of insulin resistance and diabetes in this population [26]; by excluding men receiving ADT we aimed to specifically study the effect of dysregulation of the insulin signalling pathway due to excess body fatness on PCa. Studies which had a high proportion of participants with a DM diagnosis at baseline were also excluded, although we did not exclude population-based studies which may have included some men with DM. Appendix A shows the inclusion/exclusion process we implemented for potentially eligible studies.

Exposures of interest were serum or plasma insulin or biomarkers and surrogate indices of the insulin signalling pathway and insulin resistance (fasting glucose, C-peptide, pro-insulin, HOMA-IR, HOMA-S and HbA1c).

Our outcomes of interest were PCa incidence, measures of progression (Gleason score increase, biochemical recurrence, development of local and distant metastases, change in tumour stage and decrease in number of positive cores) and PCa-specific mortality. We included any stage of cancer but investigated incident cancer and cancer progression separately. Among the included studies, a broad range of PCa definitions was reported. For the assessment of the insulin–PCa association, a set of 7 broader categories was adopted instead (see Appendix A for details).

### 2.4. Data Collection and Analysis

#### 2.4.1. Search Methods

We carried out two separate searches to identify studies examining: (i) effects of changes to body fatness on insulin signalling; and (ii) associations between biomarkers of insulin signalling and PCa outcomes. Searches were conducted using the Cochrane Database of Systematic Reviews (The Cochrane Library, July 2020; MEDLINE Ovid (from 1946 to July 2020); Embase Ovid (1980 to July 2020); and BIOSIS (1969 to July 2020). The search strategy comprised MeSH terms, text words and keywords. Full search terms that were implemented in MEDLINE, EMBASE and BIOSIS are shown in the Appendix A. Study design search filters for systematic reviews, RCTs and eligible observational studies were applied as necessary. Amendments to the search strategy were made to reflect individual database requirements. No date or language restrictions were applied. A manual search of the grey literature (Opengrey.eu (http://opengrey.eu/, accessed on 18 October 2021); Clinicaltrials.gov (https://clinicaltrials.gov/, accessed on 18 October 2021); PROSPERO (https://www.crd.york.ac.uk/PROSPERO/, accessed on 18 October 2021) was also conducted in May 2020 to source additional papers not returned in the search. Results from the literature searches were imported into Endnote X9, where duplicates were identified and removed using the Endnote function.

#### 2.4.2. Identification and Selection of Studies

We used a sequential approach to the identification and selection of studies, in terms of both the source of the studies and the types of studies (for further details, please see Appendix A).

Three reviewers (RJ, CK and LM) independently screened titles and abstracts of studies for possible inclusion against the inclusion criteria. If a title or abstract met the eligibility criteria, or eligibility could not be determined, a full-text version of the article was obtained and independently screened by two of three reviewers (RJ, CK and OD). Discrepancies between reviewers were resolved via discussion until a consensus was reached.

#### 2.4.3. Data Extraction and Management

Following the screening process, data from the eligible primary studies were independently extracted by two of three reviewers (OD, SL, SB) using a predefined data extraction form. Disagreements from this process were resolved through discussion. Data extracted for all study types included publication details (article title, year, study location), study characteristics (study design, sample size, participant demographics), intervention or exposure (setting, intervention description, how exposures measured), outcomes of interest and results (mean difference, standard deviation, *p* value, odds ratio, 95% confidence intervals). For studies of body fatness–insulin sensitivity association, we also extracted details of the intervention and the control and baseline adiposity and, within group pre–post-intervention mean differences in insulin biomarker levels; where this was not reported we calculated this ourselves as outlined in the Appendix A. For studies of insulin sensitivity- prostate cancer outcomes, we extracted information on any potential confounding factors which were adjusted for in the analysis.

#### 2.4.4. Data/Statistical Analysis

The synthesis of data was conducted separately for the two sides of the pathway. Extracted data from the included primary studies were tabulated to summarise key characteristics. We converted fasting glucose to 1 mmol/L and fasting insulin to 1 μU/mL units where these units were not already presented as such (details on the conversion factors we used are in the Appendix A).

Where we had at least 3 sufficiently similar studies (same exposures and outcomes), we performed meta-analysis of results from included studies to estimate a summary measure of effect. Both fixed-effect models and random-effects models were applied to compute pooled standardised mean difference (SMD) and relative risk (RR), with the intention of focusing on the random-effects results unless there was evidence that the model was unsuitable (e.g., evidence of small study effects which could be due to publication bias).

Results were reported and graphically displayed using forest plots where we had at least 2 studies which measured the same exposure and outcome, and which provided data in a format that allowed them to be combined. Heterogeneity in effect size between studies was assessed by estimating the between-study variance in effect sizes (τ^2^). Small study effects were assessed visually using funnel plots and tested statistically using an Egger test [27]. Where meta-analysis was not possible, results for individual studies were tabulated, data were graphically displayed using Albatross plots which are scatter plots of study sample sizes against 2-sided *p* values, allowing comparison of the direction of effect and strength of evidence across studies even when there is some heterogeneity in the exposure and outcome [28]. We also provided narrative summaries of the relevant results.

#### 2.4.5. Subgroup Analyses

For associations of body fatness with circulating insulin, we analysed studies separately according to whether the trial intervention resulted in a greater reduction in body fatness in the intervention group compared to the control group.

We also analysed studies separately by type of insulin biomarker and according to whether the PCa outcome was localised or advanced disease. We defined advanced PCa as one that had spread either to the pelvis, lymph nodes, or surrounding organs (locally advanced PCa) or PCa that had spread to more distant organs (distant metastases). High-grade PCa was defined as having a Gleason score of greater than or equal to 7 and low-grade PCa as a score less than 7. Appendix A provides details of outcome definitions by study.

### 2.5. Assessment of Methodological Quality of Included Studies

Risk of Bias of Included Studies

We included two main types of study designs within the review-RCTs and observational studies and we assessed risk of bias using selected up-to-date tools for either RCTs or non-randomised studies of exposures. We used the revised Cochrane risk-of-bias tool (RoB 2) to evaluate the risk of bias in RCTs [29]. The tool contains 5 domains, namely biases arising from: the randomisation process; deviations from intended interventions; missing outcome data; measurement of the outcome; selection of the reported result. Each study was assigned domain-level judgements and an overall judgement of risk assessed as either ‘low risk of bias’, ‘some concerns’ or ‘high risk of bias’.

For non-randomised studies of exposures, we used a preliminary version of the Risk of Bias in Non-randomised studies of Exposures (ROBINS-E) tool. Similar to ROBINS-I, bias is assessed across 7 domains, bias due to: confounding; measurement of exposure; selection of participants into the study; post-exposure interventions; missing data; measurement of outcomes; and selection of the reported result. Judgements of risk of bias are categorised as ‘low risk’, ‘low risk except for concerns of uncontrolled confounding’, ‘some concerns’, ‘high risk’ or ‘very high risk’. Results assessed to be at high’ or ‘very high’ risk of bias were excluded from syntheses.

In our inclusion criteria, we prespecified age and BMI as important confounders of the insulin–prostate cancer association and did not include studies unless they had adjusted for these confounders. Other important confounders for the insulin–PCa association were ethnicity, family history of PCa, history of cancer diagnosis (aside from nonmelanoma skin cancer), height and insulin-like growth factor (IGF) and these were used to assess risk of bias due to confounding.

Risk-of-bias assessments were performed independently by two reviewers (OD and SL), with any discrepancies resolved by discussion until a consensus was reached. An overall risk-of-bias judgement was assigned to each included study and assessments are presented in summary tables for each study.

### 2.6. Overall Assessment of the Strength of the Evidence: GRADE

The certainty in the evidence from in the included studies was rated using the GRADE framework [30], informed by the risk-of-bias status of the included studies, imprecision, heterogeneity, indirectness and reporting bias. All results started at the highest GRADE level, whether RCTs or observational studies [31]. An overall GRADE rating of high, moderate, low or very low certainty was assigned to each summary result. The assessments GRADE quality were determined by four reviewers (OD, SL, RM, JH) through discussion until a consensus was reached

## 3. Results

Our searches identified 15,478 potentially eligible studies of body fatness and insulin signalling. After removing duplicates and screening studies for eligibility, we identified seven eligible studies [32,33,34,35,36,37,38] where a reduction in body fatness was the exposure and at least one biomarker of insulin sensitivity was reported as an outcome (Figure 1).

We identified 3152 potentially eligible studies on insulin sensitivity and prostate cancer but only six of these studies (Figure 2) met our inclusion criteria [39,40,41,42,43,44].

### 3.1. Body Fatness–Insulin Association Studies

Table 1 shows the characteristics of the included studies for the body fatness–insulin association [32,33,34,35,36,37,38]. All seven eligible studies were RCTs carried out in men only, the largest of which had 80 participants and the smallest only 22 participants. The studies were carried out in Canada, Malaysia, Australia, USA, Netherlands, China, Brazil and Spain, among men with a mean age of 29 to 61. One study [32] had two separate interventions and one control group. The interventions were heterogeneous across studies but focussed on either calorie reduction or intermittent fasting.

Six of the seven studies measured fasting circulating glucose levels as the outcome, five measured fasting circulating insulin, two measured HOMA-IR, two measured both circulating glucose and insulin following a glucose tolerance test, one study measured glucose disposal rate and another one study measured circulating C-peptide.

A further 11 potential studies of interventions to reduce body fatness with biomarkers of insulin as outcomes were identified by our searches, but were subsequently excluded (Appendix A) because they were not randomised, or they did not include a placebo control group, or there was no separate analysis for men, or the study did not measure the biomarkers we were interested in, or the study was of men undergoing resistance training or a large proportion (or all) of participants were known to have pre-existing prostate cancer or metabolic syndrome.

Risk of Bias

For our body fatness–insulin association studies, we present one risk-of-bias assessment per study, because each outcome measure was based on a blood sample collected in the same manner and processed in the same laboratory. For the study by Alves et al. [32], we assessed risk of bias for each of two interventions, both aimed to reduce body weight by prescribing either a) high-carbohydrate/low-protein lunch and a high-protein/low-carbohydrate dinner or b) a high-protein/low-carbohydrate lunch and a high-carbohydrate/low-protein dinner. All seven studies of body fatness and insulin were rated as having some concerns or high risk of bias and four of these were rated as being at a high risk of bias (Appendix A). The study by Alves et al. [32] was rated as high risk of bias for both interventions. Studies were rated as high risk of bias mainly due to the high number of participants dropping out within the study and these individuals not being accounted for in the analysis.

### 3.2. Effect of Reduction in Body Fatness on Biomarkers of Insulin Sensitivity

All studies included in this systematic review achieved their aim of reducing body fatness except the study by Alves et al. [32]. All those studies in which the intervention resulted in a reduction in body fatness observed either a greater reduction or a smaller increase in insulin and glucose levels in the intervention group when comparing pre- and post-intervention levels (Appendix A and Figure 3, Figure 4 and Figure 5). The one study which achieved a reduction in body fatness, and which measured HOMA-IR [34], found a 37% reduction in levels of this biomarker post-intervention in the treatment group (from 2.64 pre-intervention to 1.67 post-intervention) but no change was seen in the control group (2.90 to 2.96), p-value for difference in pre–post-intervention change between the two groups was 0.01. Similarly, the one study to have measured C-peptide found a reduction in this biomarker following body fatness reduction in the intervention group (1.59 to 1.26 ng/L) but not in the control group (1.75 to 1.76 ng/L), *p* = 0.05. Ross et al. [37] was the only study to have measured glucose and insulin ina glucose tolerance test and to have measured glucose disposal rate following an intervention to reduce body fatness in men. This study found evidence of a faster glucose disposal rate among men in the body fatness reduction group after intervention compared with those in the control group (glucose clearance changed after intervention from 13.0 to 18.6 mg/kg muscle per minute in the intervention group compared with 15.4 to 14.4 mg/kg muscle per minute, p value for-difference between control and intervention group = 0.02).

Grade Assessment

The GRADE assessment of certainty in the evidence was downgraded by two points from high certainty to low certainty due to: the risk of bias in individual studies (all studies were at either high risk of bias or had some concerns of bias) (1 level); imprecision due to the small number of studies (0.5 level) and small number of included individuals in each study; and (0.5 level) reporting bias because these biomarkers were secondary outcomes within the RCTs we included and it is possible that other RCTs have not published data on these outcomes. We did not downgrade due to indirectness of evidence because all studies were conducted in adult men, who were recruited from the general population. We also did not downgrade the evidence due to heterogeneity as there was no evidence of this after excluding one study which did not observe a reduction in body fatness following their intervention [32].

### 3.3. Insulin–Prostate Association Cancer Studies

There were six prospective observational studies [39,40,41,42,43,44] which investigated whether biomarkers of circulating insulin were associated with prostate cancer risk at follow-up (Table 2). The studies were carried out in the USA (four studies), Iceland (1 study) and Finland (1 study), and ranged in size from 100 to 2554 men with PCa. Two of these studies measured C-peptide only, one study measured fasting glucose, another glucose tolerance, one measured fasting glucose and HbA1c and the final study measured fasting glucose, fasting insulin, the molar ratio insulin/glucose and HOMA-IR. All studies investigated the association of insulin biomarkers on total prostate cancer risk and advanced PCa, and all but one study [41] investigated localised prostate cancer.

We excluded nine studies which did not adjust their insulin biomarker-PCa results by BMI (Appendix A). The reason for this is that we were interested in the effect of body fatness on PCa only via the insulin pathway. We also excluded a further 7 studies which did not have at least 2 years follow-up between exposure and outcome measurement, 1 study which did not present data relevant to our research question, and another study which included participants who did not mean our eligibility criteria (Appendix A).

Risk of Bias

We assessed risk of bias in each study separately for each insulin biomarker exposure and each separate outcome (total PCa, localised PCa) (Appendix A). For every study and every assessment, there were some concerns relating to domain 1—confounding, due to the studies all being observational. There were also some concerns for two studies [41,44] in relation to domain 5—missing data for the outcome of total PCa. There was a high risk of bias for the same domain (5) for the advanced and high-grade PCa outcomes in the study by Dickerman et al. [41].

### 3.4. Associations between Biomarkers of Insulin Sensitivity and PCa Risk

The results for the association between insulin and PCa risk from the studies included in our review are given in Appendix A. For all exposure–outcome pairs, except fasting glucose, there were too few studies investigating the same exposure and outcome to perform meta-analyses so the results are described below in text and in the Albatross plot in Figure 6.

C-Peptide

In a small study (139 cases) published in 2010 by Lai et al. [42], high C-peptide levels were associated with a lower risk of PCa. However, in a study published in 2014 by the same authors [43], there was little evidence of an association.

Fasting Glucose and Glucose Tolerance

Three studies [39,41,44] investigated the association between fasting blood glucose levels and later risk of total and advanced PCa. The largest of these studies [41] found weak evidence that high blood glucose levels were associated with a reduced PCa risk and an effect in the same direction for advanced PCa, high-grade prostate cancer and PCa mortality, albeit with wide confidence intervals. Another study [44] found little evidence of an association with overall PCa risk and a high risk of advanced PCa and PCa mortality associated with higher blood glucose levels. The last of the three studies [39] consisted of 100 cases and did not add to the evidence base because the confidence intervals for the associations investigated in this study were very wide.

Darbinian et al. [40] investigated the effect of glucose tolerance and found that those with higher glucose levels one hour after a glucose challenge had a lower PCa risk.

HOMA-IR, Insulin and Molar Ratio

In a small study by Albanes et al. [39], the authors investigated the effect of fasting insulin, HOMA-IR and the molar ratio of insulin to glucose on PCa risk. They found that higher fasting insulin, higher HOMA-IR and a higher molar ratio of insulin to glucose were all associated with increased PCa risk, although the latter association had wider confidence intervals.

GRADE Assessment

We downgraded the evidence on insulin signalling and prostate cancer to very low certainty due to: risk of bias, which was at least moderate for all studies; heterogeneity between the studies; and imprecision (1 level for each).

## 4. Discussion

### 4.1. Overall Findings

We found very few studies (n = 7) which investigated the effect of a reduction in body fatness in males on biomarkers of insulin sensitivity. Of the eligible studies we identified, one did not observe a reduction in body fatness despite having implemented a calorie-restricted dietary intervention [32]. This same study [32] also observed that the effect of the intervention on blood biomarker levels was different to that in other studies. However, the studies which observed a reduction in body fatness as a result of their intervention, also found a subsequent decrease in blood glucose, insulin and C-peptide [33,34,35,36,37,38]. They showed that all indicators of insulin sensitivity improved following interventions to reduce body fatness.

We excluded several potentially eligible studies either because they did not carry out randomisation of intervention, they did not present their analysis separately for males, or because they were performed in men who already had impaired insulin sensitivity. The studies which were included in our review were assessed as having at least some concerns in relation to risk of bias.

It was difficult to draw conclusions on the effect of insulin sensitivity on PCa risk from the few studies we were able to include in our systematic review. With the exception of fasting glucose, there were only one or two studies investigating each biomarker and the evidence was heterogeneous. We excluded several studies either because they did not allow at least a two-year lag between exposure measurement and PCa or did not adjust their insulin biomarker-PCa association by BMI. A lag period is important to avoid reverse causation particularly in relation to PCa risk which has a long latency period. The hypothesis we were investigating in this review was whether body fatness affected PCa risk via insulin signalling. If we had included studies which did not adjust by BMI, we would be addressing the question of whether BMI was associated with PCa risk since BMI is so strongly correlated with biomarkers of insulin.

### 4.2. Strengths and Limitations of Our Review

We used methodology that we have previously developed with WCRF to systematically investigate the evidence for specific mechanisms between exposures and cancer outcomes [12]. We carried out a thorough search and employed inclusion criteria which meant that only those studies which most directly addressed our research question were included (i.e., we only included RCTs of interventions to reduce body fatness in men as our research question was whether reducing body fatness impacts on PCa risk via the insulin signalling pathway). We also sought to minimise bias by excluding studies with insufficient follow-up. We carried out risk of bias assessments on all included studies to determine the reliability of the evidence. In addition, four authors jointly carried out GRADE assessments to judge the level of certainty of the overall evidence.

However, we found that the overall certainty of the evidence was low for the effect of reducing body fatness on biomarkers of insulin sensitivity in men and very low for the effect of insulin sensitivity on PCa risk. Despite there being a wealth of evidence on the association between body fatness and insulin signalling (our searches found more than 15,478 manuscripts), there was very little evidence which met our inclusion criteria, or which directly addressed the question of whether reducing body fatness in adult men could impact this mechanistic pathway. Two important reasons for investigating mechanisms between exposures and cancer is to determine whether the association is causal or not, and to identify potential targets for intervention. The approach we have taken is more cost effective and will provide answers more quickly than a trial of interventions to reduce body fatness with cancer risk as the outcome. However, it does depend on evidence relating to the intermediate phenotype being available.

## 5. Conclusions

Our review has highlighted that insulin sensitivity is a potential mechanistic pathway via which body fatness could impact on PCa risk and has suggested that reducing body fatness may improve insulin sensitivity. However, the evidence linking insulin sensitivity to prostate cancer risk is inconclusive due to a lack of high-quality studies investigating this. Therefore, much more research is needed in this area before any firm conclusions on this mechanistic pathway can be drawn.

## Figures and Tables

**Figure 1 metabolites-11-00726-f001:**
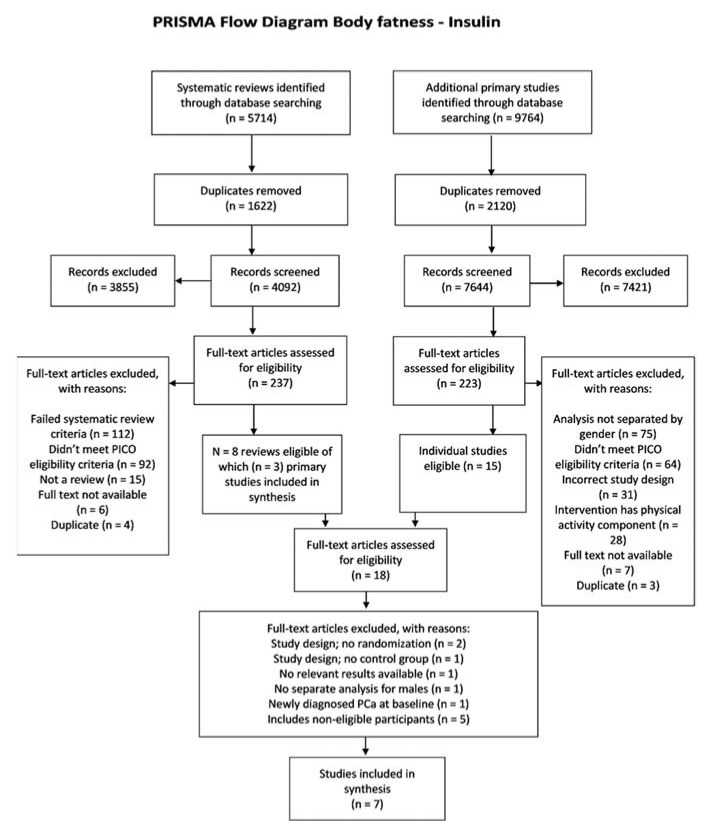
Flow diagram of body fatness–insulin association studies identified and included.

**Figure 2 metabolites-11-00726-f002:**
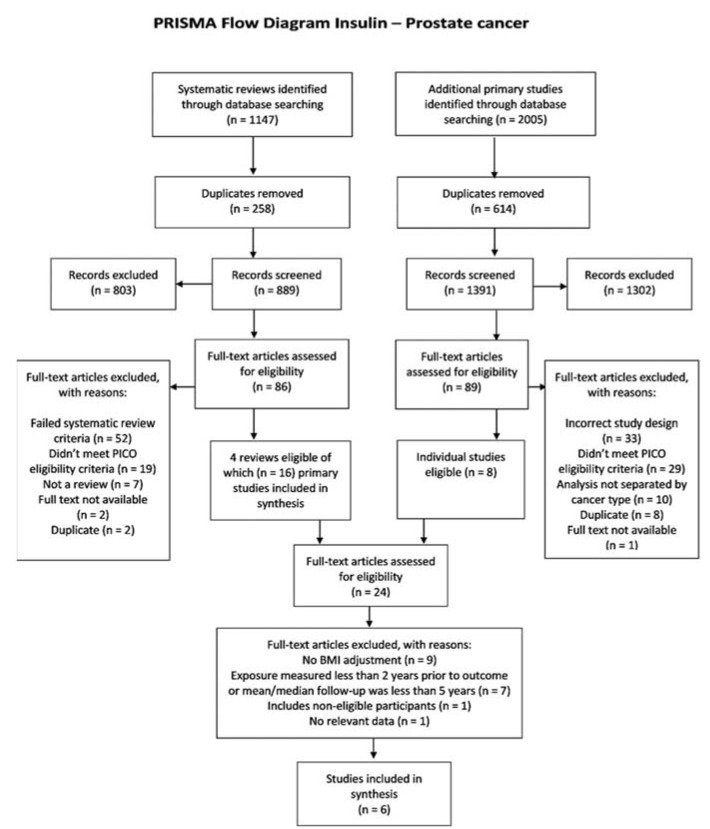
Flow diagram of insulin–prostate cancer association studies identified and included.

**Figure 3 metabolites-11-00726-f003:**
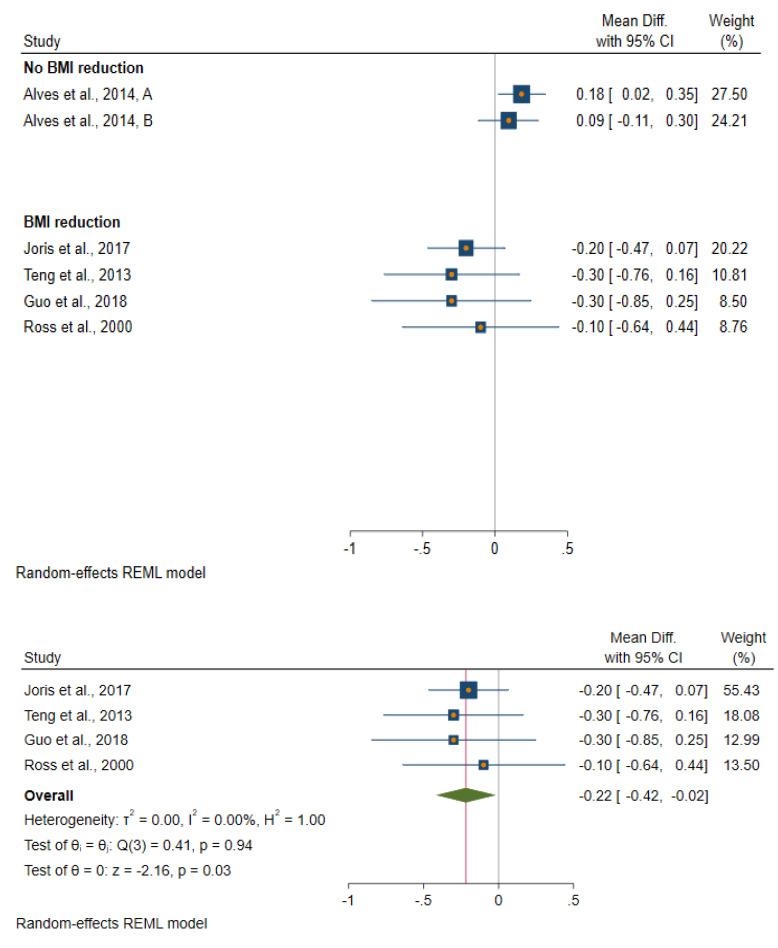
Forest plot of studies which evaluated an effect of a reduction in body fatness on fasting glucose levels.

**Figure 4 metabolites-11-00726-f004:**
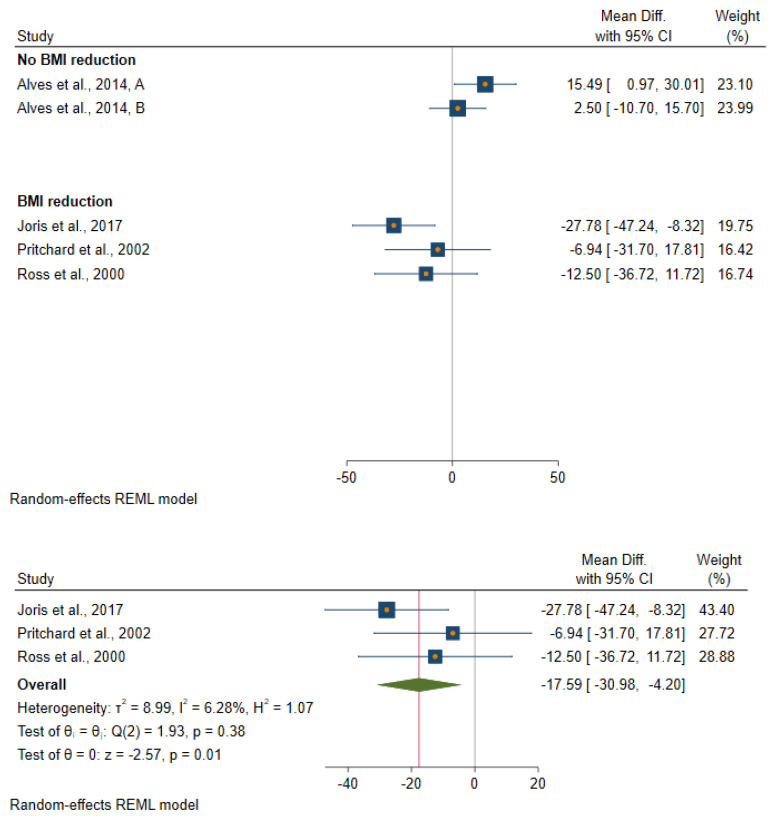
Forest plot of studies which evaluated an effect of a reduction in body fatness on fasting insulin levels.

**Figure 5 metabolites-11-00726-f005:**
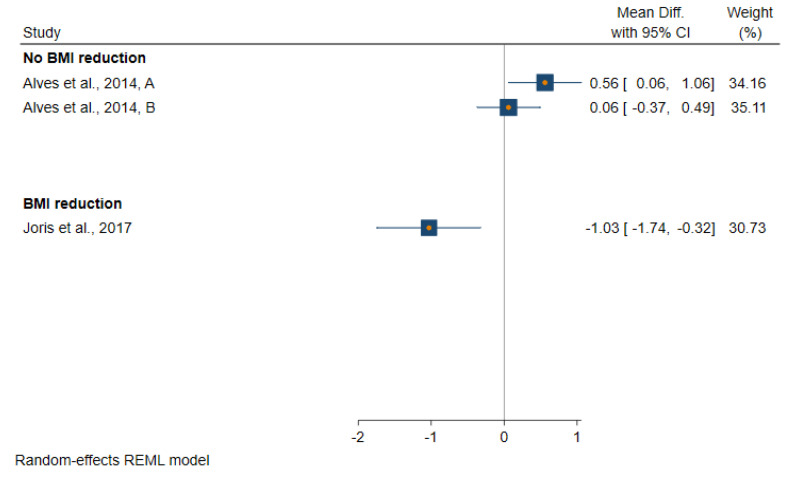
Forest plot of studies which evaluated an effect of a reduction in body fatness on HOMA-IR levels.

**Figure 6 metabolites-11-00726-f006:**
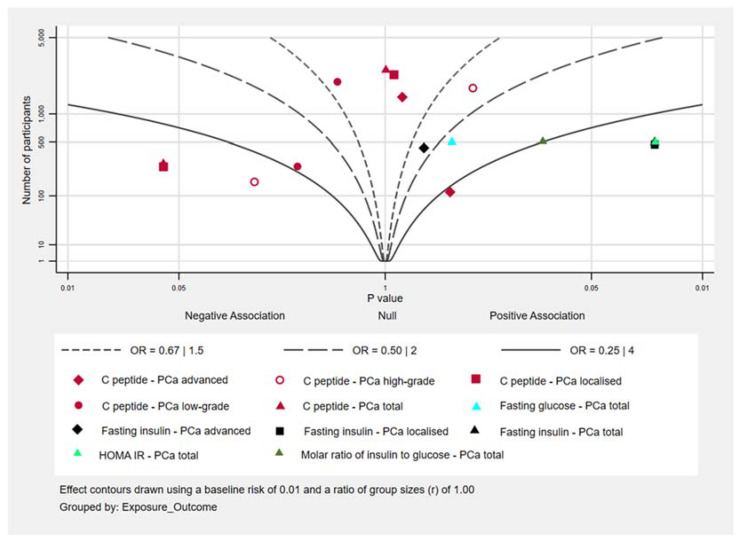
Albatross plot showing effect of insulin biomarkers on prostate cancer risk; each point on the figure represents a single analysis in an individual study.

**Table 1 metabolites-11-00726-t001:** Characteristics of included body fatness–insulin association studies.

Author (Date)	Country of Origin	Adiposity Status ^1^	Ethnicity	Number of Participants: (Intervention/Control)	Age of Participants ^2^: (Intervention/Control)	Intervention Group	Control Group	Outcome Measured
Ross R (2000)	Canada	Obese men-BMI >27 kg/m^2^	NA	22 (14/8)	42.6 (9.7)/46.0 (10.9)	Baseline period: Weight maintenance diet (4–5 weeks)		Fasting glucose
						Diet-induced weight loss group: Reduction in isocaloric diet by 700 kcal/d to achieve a weight loss of 0.6kg/week. Free-living participants (self-selected foods. Weekly 1 h seminars with dietitian. Participants kept daily food records, reviewed by the dietitian.	Body weight maintenance group: Participants asked to maintain their body weight throughout the study period. Free-living participants (self-selected foods. Weekly 1 h seminars with dietitian. Participants kept daily food records, reviewed by the dietitian.	
								Fasting insulin
								OGTT glucose (2 h)
								OGTT insulin (2 h)
								Glucose disposal rate
								Glucose disposal (Oxidative fraction)
								Glucose disposal (Nonoxidative fraction)
						Intervention period: 12 weeks		
Teng NIMF (2013)	Malaysia	BMI: 23.0- 29.9 kg/m^2^; range	Malay	56 (28/28)	59.6 (5.4)/59.1 (6.2)	Baseline period: No, but eligible subjects should not have practiced Muslim Sunnah fasting or have changed their dietary pattern three months before the study.		Fasting glucose
						Fasting calorie restriction (FCR) group: Reduction of 300–500 kcal/d combined with 2 days/week of Muslim Sunnah Fasting. During fasting day: a light meal before sunrise (Sahur), no food and drink during the day (approximately for 13 h) and a complete meal after sunset (Iftar). Subjects provided with seven-day food menu guidelines. Weekly telephone-call to obtain information regarding subjects’ dietary intake and to ensure compliance. Fasting log book and food diaries were provided during each assessment meeting.	Maintenance group: Participants were asked to maintain their present lifestyle.	
						Intervention period: 12 weeks		
Pritchard J (2002)	Australia	Overweight men-BMI: 29.0 (2.6) kg/m^2^; mean (SD)	Australian	24 [12 (10 available at baseline)/12 (2 available at baseline)]	43.4 (5.7)/43.4 (5.7)	Baseline period: No		Fasting insulin
						Low-fat (25% of dietary energy) diet group: The intervention was personalised according to the subject’s usual dietary pattern and using the National Heart Foundation booklet, The Weight Loss Guide. Compliance was monitored from food diaries and measurement of weight at monthly sessions.	Maintenance group: Participants were instructed to maintain their pre-study dietary and activity patterns, monitored at monthly measurement sessions similar to those of the intervention group.	
						Intervention period: 48 weeks		
Katzel LI (1995)	USA	Obese men-BMI: 30.0 (1.0) kg/m^2^; mean [Standard error of the mean (SEM)]	96% white (whole sample)	62 (44/18)	61.0 (1.0)/60.0 (1.0); mean [Standard error of the mean (SEM)]	Baseline period: Isoenergetic American Heart Association (AHA) phase I diet (3 months)		Fasting glucose
						Diet-induced weight loss group: Instructed to reduce energy intake by 1260 to 2100 kJ (300 to 500 kcal) per day. Goal was to decrease body weight by more than 10% during a 9 month period. Weekly group weight loss sessions. Food records were reviewed to ensure compliance to the diet.	Body weight maintenance group: Instructed not to lose weight or change their diets or level of physical activity. Weekly 1 h dietary counselling meetings to ensure compliance to the protocol.	
								Fasting insulin
								OGTT glucose (2 h)
								OGTT insulin (2 h)
						Intervention period: 36 weeks		
Joris PJ (2016)	NA; Netherlands	Abdominally obese men-Waist circumference: 102–110 cm; range	Caucasian	49 (23/26)	52.4 (46.8-61.7)/52.0 (45.4-61.1); median (Q1-Q3)	Baseline period: Measurements of abdominally obese men were balanced (18 months)		Fasting glucose
						Diet-induced weight loss group: Calorie-restricted diet for 6 weeks to obtain a waist circumference <102 cm followed by a weight-maintenance period of 2 weeks. Visited a research dietitian weekly (12 times in total) and consumed a very-low-calorie diet (VLCD) for >=4 weeks under strict guidance. If the waist circumference was still >102 cm after 4 weeks, the VLCD was continued for another week. The VLCD was supplied in powder sachets that had to be dissolved in water. Three sachets to be consumed daily. Participants were allowed to eat 250 g vegetables or fruit/day. After the VLCD period, subjects were prescribed a mixed, solid, calorie-restricted diet.	Body weight maintenance group: Maintained their habitual diet, physical activity levels, and use of alcohol throughout the total study period. Visited a research dietitian on 2 occasions.	
								Fasting insulin
								C-peptide
								HOMA-IR
						Intervention period: 8 weeks (a calorie-restricted diet for 6 weeks to obtain a waist circumference <102 cm followed by a weight-maintenance period of 2 weeks)		
Guo X (2018)	China	Overweight/obese men-BMI > 24 kg/m^2^	Chinese	80 (42/38)	38.9 (6.5)/38.0 (6.6)	Baseline period: No		Fasting glucose
						Meal replacement with mild caloric restriction group: Consumed one liquid meal replacement which contained 388 kcal in total energy at dinner time during the intervention. Individuals were advised to continue their regular physical activity regimen. Dietary habits were assessed through a self-administered 77-item Food Frequency Questionnaire (FFQ) at the first and last visit (12th week).	Routine diet group: Followed a routine Chinese dinner as before. Individuals were advised to continue their regular physical activity regimen.	
						Intervention period: 12 weeks		
Alves RDM (2014)	Brazil and Spain	Overweight/obese men-BMI: 30.1 (2.8) kg/m^2^; mean (SD)	NA	39 (21/18)	29.3 (7.3)/31.4 (7.6)	Baseline period: Weight-maintaining diet (3 days)		Fasting glucose
						Hypocaloric diet (~10% of caloric restriction)-Diurnal carbohydrate/nocturnal protein (DCNP) group: Received a prescription of a high-carbohydrate/low-protein lunch (69.3 and 7.2%, respectively) and a high-protein/low-carbohydrate dinner (41.7 and 18.8%, respectively). Subjects were asked to maintain habitual physical activity. Subject received nutritional advice and education from registered dietitians. Instructed to use an exchange-based self-selected food list, which assigned foods into categories according to their macronutrient composition. Subjects provided two 3-day food records (2 week days and 1 weekend day	Macronutrient-balanced group: Macronutrient-balanced lunch and dinner (18.0% protein, 46.8% carbohydrate, 35.2% fat).	
								Fasting insulin
								HOMA-IR
						Intervention period: 8 weeks		
Alves RDM (2014)	Brazil and Spain	Overweight/obese men-BMI: 30.1 (2.8) kg/m^2^; mean (SD)	NA	37 (19/18)	29.5 (7.5)/31.4 (7.6)	As Alves et al. (2014a) above except the lunch and dinner in the intervention group were reversed.		

^1^ BMI unless otherwise stated, ^2^ mean (SD) unless otherwise specified.

**Table 2 metabolites-11-00726-t002:** Characteristics of included studies (insulin–PCa association set of studies).

Case–Control Studies Nested in a Prospective Cohort										
Author (Date)	Country of Origin	Study Name	Source of Participants	Duration of Follow-Up ^1^	Ethnicity	Number of Participants (Cases/Controls)	Age ^2^ of Participants at Baseline (Cases/Controls)	Exposure Measured	Outcomes Assessed	Adjustment Variables
Lai GY (2010)	USA	CLUE II cohort	General population	5.6 years (mean), (range: 0.3–12.1 years)	Majority White Americans; 2.3% African Americans (cases and controls)	139/139	64.6 (9.0)/64.6 (9.0)	C-peptide	PCa total	BMI (overweight: 25-29.9, obese: ≥30, normal: <25 kg/m^2^), family history of prostate cancer (yes, missing, no)
						127/127			PCa, localised	
						57/57			PCa, advanced	
						128/128			PCa, low-grade	
						80/80			PCa, high-grade	
Lai GY (2014)	USA	Health Professionals Follow-Up Study (HPFS)	Occupational group (health professionals)	5.4 years (median) (IQR: 3.1–7.7 years)	White Americans (cases: 94.2%, controls: 92.9%)	1314/1314	64.2 (40.0-75.0)/64.2 (40.0-75.0); mean (range)	C-peptide	PCa total	BMI (kg/m^2^, continuous), history of diabetes
						1064/1314			PCa, localised	BMI (kg/m^2^, continuous), history of diabetes, height (in, continuous), first degree family history of prostate cancer, vigorous physical activity (MET-hrs/wk, continuous), smoking in the past 10 years, history of vasectomy, total energy intake (kcal/day, continuous), alcohol intake (g/day), energy-adjusted intake of calcium (mg/day), alpha-linolenic acid (g/day), lycopene (μg/day), fructose (g/day), cumulative updated intake (1986–1994) of red meat and fish (servings/week), use of a vitamin E or selenium supplement
						156/1314			PCa, advanced	
						736/1314			PCa, low-grade	
						477/1314			PCa, high-grade	
Albanes D (2009)	Finland	Alpha-Tocopherol, Beta-Carotene Cancer Prevention (ATBC) Study	General population	9.2 years (mean), (range: 5–12 years)	Finnish	100/400	59.0 (4.6)/56.4 (5.0)	Fasting insulin	PCa total	Age (years), BMI (kg/m^2^)
						69/400			PCa, localised	
						30/400			PCa, advanced	
						100/400		Fasting glucose	PCa total	
						100/400		Molar ratio of insulin to glucose		
						100/400		HOMA-IR		
Prospective cohorts										
**Author (Date)**	**Country of Origin**	**Study Name**	**Source of Participants**	**Duration of Follow-Up**	**Ethnicity**	**Number of Participants (Cases/Total)**	**Age of Participants at Baseline**	**Exposure Measured**	**Outcomes Assessed**	**Adjusted Variables**
						152/1492		HOMA-IR		
						75/1215		HbA1c (%)		
						152/1,493		Fasting insulin		
Dickerman BA (2018)	Iceland	The Reykjavik Study	General population	25 years (mean)	Icelandic	1061/9097	52.0; median	Fasting glucose	PCa total	Entry age (linear and quadratic terms) and stage (categorical) of cohort entry (1967–68, 1970–71, 1974–76, 1979–81, 1985–87), family history of prostate cancer (yes, no), smoking status (never, former, current), regular check-ups (yes, no), attained education (primary, secondary, college, university), height (quartiles), BMI (<25.0, 25.0–29.9, ≥30 kg/m^2^)
						374/9097			PCa, high-grade	
						145/9097			PCa, advanced	
						336/9097			PCa mortality	
Marrone MT (2019)	USA	The Atherosclerosis Risk in Communities (ARIC) Study	General population	22 years (max)	27% African American	626/4127	48.0–67.0; range	Fasting glucose	PCa total	Age (continuous, visit 2), joint categories for race and field centre (White from Minnesota; White from Washington Co. or Forsyth Co.; Black from Jackson; Black from Washington Co. or Forsyth Co.), BMI (kg/m^2^, continuous, visit 2), waist circumference (cm, continuous, visit 2), education (<high school, high school with some college, college graduate), cigarette smoking status (current/former smoker who quit <10 years ago; former smoker who quit ≥10 years ago, never smoker, visit 2)
						64/4689			PCa, advanced	
						59/4694			PCa mortality	
						626/4127		HbA1c (%)	PCa total	
						64/4689			PCa, advanced	
						59/4694			PCa mortality	
Darbinian JA (2008)	NA; USA	Kaiser Permanente Medical Care Program	General population	18.4 years (median)	White: 78.6%; Black: 13.2%; Asian: 4.4%; Other: 3.8%	2554/*	48.0 (35.0–80.0); median (range)	Glucose tolerance	PCa total	Glycaemic status (serum glucose levels measured 1 h after ingestion of 75 g oral glucose challenge among MHC examination participants who did not self-report history of diabetes or as diabetes per self-report (at MHC examination) of either physician diagnosis or diabetes-related medication usage during past year or two), year of MHC examination (<55, ≥55), race/ethnicity (White, African American), BMI per the WHO classification (<25, ≥25 kg/m^2^)
						1727/*			PCa, localised	
						642/*			PCa, regional (stages 2–5), distant (stage 7)	

^1^ Time from measurement of exposure until outcome diagnosis. ^2^ Mean (standard deviation) unless otherwise stated. * At the time of each outcome, a risk set was formed in which the person diagnosed with cancer was compared with all others in the cohort who were within 2–4 years of calendar year of birth(full cohort: 47,209 participants) and adjusted for year of examination. For ease of reporting we have separated nested case-control studies from full cohort studies (labelled prospective cohorts) although both study types are prospective in design.

## Data Availability

The data presented in this study are taken from published sources and are available in the article and Appendix A.

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
