# Peer review of "Could Reducing Body Fatness Reduce the Risk of Aggressive Prostate Cancer via the Insulin Signalling Pathway? A Systematic Review of the Mechanistic Pathway"

_metabolites, 2021, doi:10.3390/metabo11110726_

Round 1

Reviewer 1 Report

In this review, the authors claim to have insulin sensitivity is a potential mechanistic path- 456 way via which body fatness could impact on PCa risk and has suggested that reducing 457 body fatness may improve insulin sensitivity.

Author Response

No comments to address were provided.

Reviewer 2 Report

The authors present a paper about "Could reducing body fatness reduce the risk of aggressive prostate cancer via the insulin signalling pathway? A systematic review of the mechanistic pathway".

The topic is interesting and the authors have certainly made a great effort to provide the final result.

I have some concersn which I would the authors to address:

1) It is not clear the contraditcion between the statement made in the abstrct "We were unable to reach any conclusions on the effect of insulin sensitivity on PCa risk from the few studies included in our systematic review" and what authors state in the conclusion of the review "Our review has highlighted that insulin sensitivity is a potential mechanistic pathway via which body fatness could impact on PCa risk". 

Please explain which is the real finding of your review

2) The authors state "In this systematic review, we use the two-step methodology we previously developed".

Please do not give it for granted that all the researchers know your previous works so do not just cite your previous paper but explain this double approach more clearly.

3) please correct some typos scattered in the text (such as double spaces)

Author Response

1) It is not clear the contradiction between the statement made in the abstract "We were unable to reach any conclusions on the effect of insulin sensitivity on PCa risk from the few studies included in our systematic review" and what authors state in the conclusion of the review "Our review has highlighted that insulin sensitivity is a potential mechanistic pathway via which body fatness could impact on PCa risk". Therefore Please explain which is the real finding of your review

Response: In our review we found some evidence that reducing body fatness improves insulin sensitivity in men, therefore this is a potential mechanism via which body fatness could impact on PCa risk. However, we were unable to conclude whether insulin sensitivity influences prostate cancer risk because there were very few high quality studies which had investigated this, although we did not find any evidence to the contrary. We have changed the wording of the conclusion to the text below to make this clearer.

Our review has highlighted that insulin sensitivity is a potential mechanistic pathway via which body fatness could impact on PCa risk and has suggested that reducing body fatness may improve insulin sensitivity. However, the evidence linking insulin sensitivity to prostate cancer risk is inconclusive due to a lack of high quality studies investigating this. Therefore much more research is needed in this area before any firm conclusions on this mechanistic pathway can be drawn.”

2) The authors state "In this systematic review, we use the two-step methodology we previously developed".

Please do not give it for granted that all the researchers know your previous works so do not just cite your previous paper but explain this double approach more clearly.

Response: We  have now added the following sentences to explain this methodology:

In this methodology the first stage is designed to identify mechanisms underpinning a specific exposure–disease relationship and prioritise these mechanisms using specifically designed text mining tools. The second stage is a targeted systematic review of studies of the exposure of interest and intermediate phenotypes relating to the mechanism of interest, and a separate systematic review on studies on the same intermediates and outcome of interest.”

3) please correct some typos scattered in the text (such as double spaces)

We have edited the manuscript and corrected any typos we have spotted.

Reviewer 3 Report

A NICE WORK!

Author Response

No comments to address.